# ALLIES: Prompting Large Language Model with Beam Search

**Hao Sun**[1][*]**, Xiao Liu**[2][†]**, Yeyun Gong**[2]**, Yan Zhang**[1]**, Daxin Jiang**[3]**, Linjun Yang**[3]**, Nan Duan**[2]

[1] Peking University, [2] Microsoft Research Asia, [3] Microsoft

sunhao@stu.pku.edu.cn, zhyzhy001@pku.edu.cn,

{xiaoliu2,yegong,nanduan}@microsoft.com

## Abstract

With the advance of large language models (LLMs), the research field of LLM applications becomes more and more popular and the idea of constructing pipelines to accomplish complex tasks by stacking LLM API calls come true. However, this kind of methods face two limitations: narrow information coverage and low fault tolerance. In this work, we propose a novel method called ALLIES. Given an input query, ALLIES leverages LLMs to iteratively generate new queries related to the original query, enabling an iterative reasoning process. By iteratively refining and expanding the scope of the original query, ALLIES captures and utilizes hidden knowledge that may not be directly obtainable through retrieval. We take zero-shot open-domain question answering (ODQA) as an application scene and evaluate ALLIES on the widely-used benchmarks, such as NQ, WebQ and TriviaQA. The experimental results demonstrate that ALLIES significantly outperforms other zero-shot baselines, indicating its effectiveness in tackling those challenges. Our code is available in https://github.com/microsoft/SimXNS/tree/main/ALLIES.

## 1 Introduction

With the emergence of large language models (LLMs) [OpenAI, 2023, Scao et al., 2022, Taylor et al., 2022, Chowdhery et al., 2022], researchers have explored their potential to generate responses, including answering queries with the in-context learning method [Brown et al., 2020]. In that method, the models are prompted with demonstrations such as human-selected query-response pairs [Shoeybi et al., 2019, Rae et al., 2021, Du et al., 2022]. In this field, open-domain question answering [Chen et al., 2017, Izacard and Grave, 2021, 2020, Lazaridou et al., 2022] is an important and representative task that usually requires

---

*This work was done during internship at MSRA.

†Xiao Liu is the corresponding author.

access to external corpora [Petroni et al., 2021] and utilizes a retriever component for knowledge augmentation [Ram et al., 2023, Shi et al., 2023, Rashkin et al., 2021, Gao et al., 2022, Bohnet et al., 2022, Menick et al., 2022] to improve their ability to provide comprehensive and accurate answers.

However, despite the advancements, these methods still face two main limitations. (1) Firstly, **narrow information coverage**. When incorporating relevant information, the majority of these approaches only employ the query itself to find or retrieve additional contextual information. Nonetheless, there are instances where responding to the query necessitates implicit knowledge that is related to the query but cannot be easily found solely using the given query. Consequently, the LLM may fail to acquire crucial information required for accurately responding to the query. (2) Secondly, **low fault tolerance**. Most of these methods follow the pipeline style, consisting of unique steps calling LLM APIs to generate responses to fulfill different needs in a single turn. It means that the model is expected to provide the correct response in a single attempt. If an internal step fails, either the whole pipeline will face the risk of exception or the error will be propagated to downstream steps. Consequently, if the model fails to find the necessary information or misinterprets the question, it may produce an incorrect response.

To address the aforementioned limitations, we propose a novel approach called ALLIES that applies a beam search strategy to generate responses. To better elaborate the method, we take open-domain question answering as the application scene and show an example of how ALLIES works in Figure 1. We adopt an interactive and iterative process. Initially, we generate additional queries by asking the LLM what other information they require, based on the existing query-evidence pair. These generated queries serve as prompts for retrieving relevant evidence from external sources.

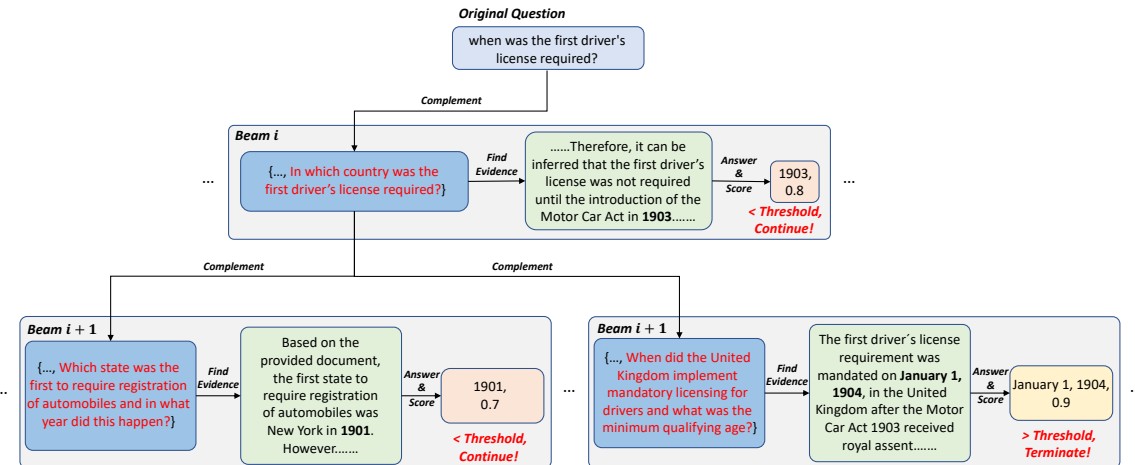

Figure 1: The example of answering a question "*when was the first driver's license required?*" using ALLIES. The correct answer is "*January 1, 1904*".

The retrieved evidence is then added to the existing query-evidence pair. Next, we employ the LLM to respond to the initial query based on the augmented query-evidence pairs. Subsequently, we solicit the LLM to score the response, taking into account the query and the augmented query-evidence pair. This scoring process provides a measure of confidence in the generated response. The iterations continue until the score surpasses a predefined threshold, indicating a sufficiently confident answer or the maximum depth of the tree traversal is reached. Once either of these conditions is fulfilled, the process terminates, and the answer is outputted as the final result. Responding to the query using ALLIES can be conceptualized as a tree traversal process, starting from the root node and progressing towards the leaf nodes, where each internal node in the tree represents a generated query.

The main advantages of our method are two folds: (1) Firstly, we employ an extension strategy that extends the original question to multiple relevant questions, broadening the information coverage. This approach enables the LLM to gain a deeper understanding of the complex question by focusing on its constituent parts. By providing the LLM with more specific and targeted queries, we enhance their ability to comprehend and process the question effectively. (2) Secondly, during the iterative process, we employ a dynamic pruning technique that retains only the top $B$ answers at each step. This increases the fault tolerance and robustness of our model by allowing the LLM to

make mistakes during the reasoning process. Any erroneous answers can be replaced by alternative answers, leading to more accurate and reliable responses. This flexibility and adaptability contribute to the improved performance of our approach.

With the idea of ALLIES, we take zero-shot open-domain question answering (ODQA) as an application scene and evaluate ALLIES in several popular benchmarks. We conduct experiments on the NQ, TriviaQA and WebQ datasets. The results demonstrate that ALLIES significantly outperforms several representative baselines while maintaining an acceptable cost. The case study further confirms the aforementioned advantages of our method.

In summary, our main contributions can be summarized as follows:

1. We propose ALLIES, which leverages a beam search strategy for response generation. Within this framework, we adopt an interactive and iterative process to enhance the accuracy and robustness of the responses.

2. By extending the original question into multiple relevant questions and employing a dynamic pruning technique, we improve the understanding of complex questions and increase the model's robustness. This allows for mistakes and alternative answers, resulting in more accurate and robust responses.

3. By taking zero-shot ODQA as an application scene, results on the NQ, TriviaQA and WebQ

datasets demonstrate the significant outperformance of our method compared to baseline approaches. The case study further validates the advantages of our approach.

## 2    Related Work

### 2.1    Open-Domain Question Answering

Open-domain question answering is a task that aims to provide answers to questions without relying on specific context. This task can be categorized into two settings: the open-book setting and the closed-book setting. In the open-book setting, models [Chen et al., 2017, Izacard and Grave, 2021, 2020] typically consist of a retriever and a reader component. The retriever's role is to retrieve relevant information from a corpus such as Wikipedia [Chen et al., 2017, Izacard and Grave, 2021] or web pages [Lazaridou et al., 2022, Nakano et al., 2021], while the reader focuses on answering the question based on the retrieved information.

In the closed-book setting, models have no access to external corpus and have to rely on its model parameters to store all the information. Recent works find that large-scale language models like T5 [Raffel et al., 2020] can already answer questions without access to the external corpus. However, small-scale language models like RoBERTa [Liu et al., 2019] or GPT-2 [Radford et al., 2019] still face challenges in accurately answering questions in this setting.

### 2.2    Large Language Model Enhanced Question Answering

In recent times, there has been a shift towards utilizing large language models (LLMs) for question answering [Chowdhery et al., 2022, Du et al., 2022, Liu et al., 2021]. This research can be broadly categorized into two lines of work. The first line of work focuses on preprocess methods [Borgeaud et al., 2022, Ram et al., 2023, Shi et al., 2023], which involve obtaining relevant documents and then utilizing LLMs to generate answers. Within this line of work, there are two main approaches. Retrieve-then-read methods [Ram et al., 2023, Shi et al., 2023] employ a retrieval model to retrieve relevant documents, while generate-then-read methods [Yu et al., 2022, Sun et al., 2022] fully leverage the capabilities of LLMs. Furthermore, researchers have demonstrated that combining generation and retrieval can lead to further gains [Yu et al., 2022].

The second line focuses on posthoc methods

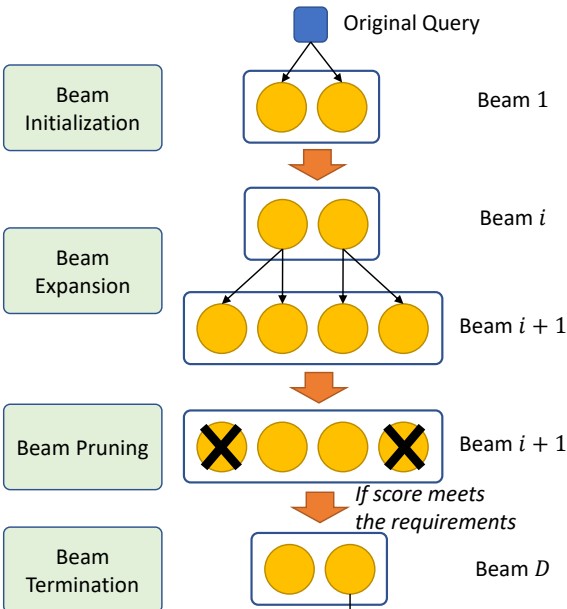

Figure 2: The abstract process of ALLIES.

(like works on QA with attribution) [Rashkin et al., 2021, Gao et al., 2022, Bohnet et al., 2022, Menick et al., 2022], which involve generating an answer using an LLM and then refining it with the help of a verifier and a retriever. The retrieved documents in the second stage serve as explanations for the generated answer.

## 3    Main Idea

The main idea of ALLIES is an interactive and iterative process based on the widely-used search algorithm, beam search[1]. We use a tuple with five slots to represent a *state*, which is the element of a *beam*. Each *state* $\langle q, \mathcal{Q}, \mathcal{E}, r, s \rangle$ consists of the original query $q$, the set of historical query completions $\mathcal{Q}$, the set of historical external evidences $\mathcal{E}$, the current response $r$, and the estimated score $s$ according to the current state. Assume the maximum search depth is $D$, as illustrated in Figure 2, there are four main stages of ALLIES.

### 3.1    Beam Initialization

In the beginning, we initialize the beam by asking the LLM to answer the query directly and by answering the query based on retrieved evidence. The retrieved evidence is obtained by first retrieving related documents using the original query and then summarizing the documents. The generated tuples will be added to the beam.

---

[1]https://archive.org/details/DTIC_ADA049288

**Algorithm 1** The process of generating the response to a given query using ALLIES.

**Hyperparameters**: The maximum number $K$ of generated queries, the maximum depth $D$ of extension, the number $N$ of documents from retrieval, the score threshold $S$, and the beam size $B$.
**Input**: A query $q$.
**Output**: The answer $\hat{a}$.

1: Clear the initial beam $\mathcal{S}_0 = \varnothing$
2: Answer the query $q$ with the model knowledge $a_0 = \text{Answer}(q, \varnothing, \varnothing)$.
3: Score the initial answer $s_0 = \text{Score}(q, \varnothing, \varnothing, a_0)$.
4: Add the current tuple to the initial beam $\mathcal{S}_0 = \mathcal{S}_0 \cup \{\langle q, \varnothing, \varnothing, a_0, s_0 \rangle\}$.                    ▷ The first seed.
5: Retrieve a evidence $e_1 = \text{Retrieve}(q_{ori}, q, N)$.
6: Answer the query $q$ with the model knowledge $a_1 = \text{Answer}(q, \{q\}, \{e_1\})$.
7: Score the initial answer $s_1 = \text{Score}(q, \{q\}, \{e_1\}, a_1)$.
8: Add the current tuple to the initial beam $\mathcal{S}_0 = \mathcal{S}_0 \cup \{\langle q, \{q\}, \{e_1\}, a_1, s_1 \rangle\}$.             ▷ The second seed.
9: **for** extension depth $d$ in $1 \rightarrow D$ **do**                                                      ▷ Extending within the depth.
10:     Clear the beam for the current depth $\mathcal{S}_d = \varnothing$.
11:     **for** each tuple in the previous beam $\langle q, \mathcal{Q}, \mathcal{E}, a, s \rangle \in \mathcal{S}_{d-1}$ **do**                    ▷ Iterate the previous tuples.
12:         Find the extended queries $\mathcal{Q}' = \text{Ask}(q, \mathcal{Q}, \mathcal{E}, K)$.
13:         **for** each extended query $q' \in \mathcal{Q}'$ **do**                                            ▷ Try each possible extension.
14:             Retrieve a evidence $e' = \text{Retrieve}(q_{ori}, q', N)$.
15:             Try to answer with all the evidences $a' = \text{Answer}(q, \mathcal{Q} \cup \{q'\}, \mathcal{E} \cup \{e'\})$.
16:             Score the answer $s' = \text{Score}(q, \mathcal{Q} \cup \{q'\}, \mathcal{E} \cup \{e'\}, a')$.
17:             Add the current extended tuple to the beam $\mathcal{S}_d = \mathcal{S}_d \cup \{\langle q, \mathcal{Q} \cup \{q'\}, \mathcal{E} \cup \{e'\}, a', s' \rangle\}$.
18:         **end for**
19:     **end for**
20:     Trim the beam $\mathcal{S}_d$ by keeping only $B$ tuples with largerest scores.                         ▷ Prune the beam.
21:     **if** a tuple $\langle q, \mathcal{Q}, \mathcal{E}, a, s \rangle \in \mathcal{S}_d$ meets $s \geq S$ **then**                         ▷ Examine the exit.
22:         $\mathcal{S}_D = \mathcal{S}_d$.
23:         Exit the loop.
24:     **end if**
25: **end for**
26: Find the tuple $\langle q, \mathcal{Q}, \mathcal{E}, \hat{a}, s_{\max} \rangle \in \mathcal{S}_D$ with the largest score $s_{\max}$ and $\hat{a}$ is the final answer.

## 3.2 Beam Expansion

During the beam search process, we iteratively pop out one element from the front of the beam. For each element, we generate queries using the Ask Function. Then, for each generated query, we retrieve relevant evidence and ask the LLM to answer the query based on both the retrieved evidence and the reasoning history. The LLM scores the generated answers based on the reasoning history, and the newly formatted tuples are added to the end of the beam.

## 3.3 Beam Pruning

At the end of each search depth, we rank the newly generated answers and keep only top $B$ answers.

## 3.4 Beam Termination

If the highest-ranking answer in the beam has a score exceeding the predefined threshold, the search process terminates, and the answer is outputted. Otherwise, the process continues. If none of the elements in the beam reaches the threshold, we output the highest-scoring answer when the search reaches the maximum depth.

## 4 Detailed Approach for ODQA

In this section, we present the application of AL-LIES in ODQA, whose algorithm is illustrated in Algorithm 1. There are four key functions used in ALLIES, each serving a specific purpose. The corresponding prompts are illustrated in Appendix C.

### 4.1 Answering Function $\text{Answer}(q, \mathcal{Q}, \mathcal{E})$

This function takes as input the original query $q$, previously generated queries $\mathcal{Q}$, and corresponding retrieval evidence $\mathcal{E}$. It constructs a reasoning history $\{\langle q_1, e_1 \rangle, \langle q_2, e_2 \rangle, ...\}$ by extracting $q_i \in \mathcal{Q}$ and $e_i \in \mathcal{E}$. The function then asks the LLM to reason over the reasoning history and provide an answer to the original query.

### 4.2 Asking Function $\text{Ask}(q, \mathcal{Q}, \mathcal{E}, K)$

Given the query $q$, previously generated queries $\mathcal{Q}$, corresponding retrieval evidence $\mathcal{E}$, and the maximum number of queries to be generated $K$, this function constructs a reasoning history $\{\langle q_1, e_1 \rangle, \langle q_2, e_2 \rangle, ...\}$ by extracting $q_i \in \mathcal{Q}$ and $e_i \in \mathcal{E}$. The LLM is then asked to reason over the reasoning history and determine what additional information it requires to answer the question. The function outputs the generated queries.

### 4.3 Retrieval Function $\text{Retrieve}(q_{ori}, q, N)$

Given the original query $q_{ori}$, query $q$, and the maximum number of documents to be retrieved $N$, this function uses a dense retriever to retrieve the top-$N$ most similar documents. The LLM is then asked to extract the most useful information from the documents and summarize them, providing a concise version of the retrieved information. We can also use LLM to directly generate a background document like GENREAD [Yu et al., 2022] as an alternative and we call this function $\text{Retrieve}'(q_{ori})$.

### 4.4 Scoring Function $\text{Score}(q, \mathcal{Q}, \mathcal{E}, a)$

Given the original query $q$, previously generated queries $\mathcal{Q}$, corresponding retrieval evidence $\mathcal{E}$, and the generated answer $a$ from the LLM, this function constructs a reasoning history $\{\langle q_1, e_1 \rangle, \langle q_2, e_2 \rangle, ...\}$ by extracting $q_i \in \mathcal{Q}$ and $e_i \in \mathcal{E}$. The LLM is then asked to consider the reasoning history and assess the probability that the candidate answer is the true answer. The function outputs a score representing the confidence in the generated answer.

## 5 Experiment

### 5.1 Experimental Setting

In this section, we conduct experiments on three open-domain question-answering (QA) datasets: NQ [Kwiatkowski et al., 2019], TriviaQA [Joshi et al., 2017], and WebQ [Berant et al., 2013]. Since we focus on zero-shot ODQA, we utilize only the complete test sets of NQ and WebQ. To reduce costs, we randomly selected 1000 samples from the TriviaQA test set for evaluation purposes. Original detailed statistics regarding these three datasets can be found in Appendix A.

We evaluate the performance using two metrics: the exact match (EM) score and the F1 score. Specifically, a predicted answer is considered correct only if its normalized form matches any of the normalized versions of the answers provided in the answer list. The F1 score measures the word overlap between the normalized version of the predicted answer and the answers in the provided answer list.

### 5.2 Implementation

We employ GPT-3.5-Turbo hosted by Azure OpenAI services as our large language model (LLM). As for the retriever component, we conduct separate finetuning for the NQ, TriviaQA, and WebQ datasets using their respective training sets. The architecture and performance of the dense retrieval component can be found in Appendix D. For the retrieval corpus, we use the Wikipedia dump from Dec. 20, 2018 as our retrieval corpus, encompassing a collection of 21,015,324 documents.

### 5.3 Baselines

We compare our method with three groups of zero-shot QA baselines.

The first group comprises baselines that utilize a retriever in their approach. This includes models such as BM25 + InstructGPT, Contriever + InstructGPT, Google + InstructGPT, and DPR + InstructGPT. These models employ a retriever to retrieve relevant information, which is then used by InstructGPT for answer generation. We obtained the reported performance numbers for these baselines from GENREAD [Yu et al., 2022].

The second group consists of baselines that do not utilize a retriever in their approach. This group includes models such as GPT-3 [Brown et al., 2020], InstructGPT [Yu et al., 2022], FLAN [Wei et al., 2021], GLaM [Du et al., 2022], and GENREAD [Yu et al., 2022]. The reported performance numbers for these baselines are obtained from their respective original papers.

The third group consists of models that we implemented ourselves, including directly answer, retrieve-then-answer, GENREAD [Yu et al., 2022], self-Ask [Press et al., 2022], and MCR [Yoran et al., 2023]. Directly answer refers to the utilization of the LLM to directly answer the question. Retrieve-then-answer involves retrieval before answering, where we experimented with different numbers of retrieved documents and reported their corresponding performance, which is the simplified version of ALLIES without beam search. We implemented GENREAD, self-Ask, and MCR based on their open-source code. However, we evaluate MCR only on the NQ dataset due to its high API cost. To ensure fairness among the baselines, we set the retrievers and LLM configurations to be the same.

### 5.4 Main Results

We present the main results of our zero-shot experiments in Table 1. Based on these results, several observations can be made:

(1) Among the methods that utilize a retriever, the choice of the retriever has a significant impact on the model's performance. This indicates that the quality of the retrieved documents plays a crucial role in determining the overall system performance.

| Method | NQ | | TriviaQA | | WebQ | |
|--------|-----|-----|-----|-----|-----|-----|
| | EM | F1 | EM | F1 | EM | F1 |
| *Method w/ retriever, reported by [Yu et al., 2022].* | | | | | | |
| BM25 + InstructGPT | 19.7 | - | 52.2 | - | 15.8 | - |
| Contriever + InstructGPT | 18.0 | - | 51.3 | - | 16.6 | - |
| Google + InstructGPT | 28.8 | - | 58.8 | - | 20.4 | - |
| DPR + InstructGPT | 29.1 | - | 53.8 | - | 20.2 | - |
| *Method w/o retriever.* | | | | | | |
| GPT-3 [Brown et al., 2020] | 14.6 | - | - | - | 14.4 | - |
| InstructGPT [Yu et al., 2022] | 20.9 | - | 57.5 | - | 18.6 | - |
| FLAN [Wei et al., 2021] | 18.6 | - | 55.0 | - | - | - |
| GLaM [Du et al., 2022] | 24.7 | - | - | - | 19.0 | - |
| *Reimplmentation.* | | | | | | |
| Directly Answer | 20.8 | 32.5 | 49.2 | 60.8 | 20.8 | 37.5 |
| Retrieve-Then-Answer (Top-1) | 27.6 | 37.1 | 49.1 | 57.9 | 19.9 | 33.8 |
| Retrieve-Then-Answer (Top-5) | 29.4 | 40.7 | 52.7 | 62.0 | 18.5 | 34.8 |
| Retrieve-Then-Answer (Top-10) | 28.2 | 39.5 | 52.4 | 61.6 | 17.4 | 32.9 |
| GENREAD [Yu et al., 2022] | 31.1 | 44.8 | 59.3 | 70.7 | 19.1 | 36.9 |
| Self-Ask [Press et al., 2022] | 26.4 | 36.5 | 59.4 | 68.5 | 15.1 | 29.5 |
| MCR [Yoran et al., 2023] | 27.1 | 35.7 | - | - | - | - |
| ALLIES | **38.0** | **47.8** | **61.4** | **70.8** | **28.2** | **45.6** |

Table 1: Zero-shot open-domain QA performance.

| Method | NQ | | WebQ | |
|--------|-----|-----|-----|-----|
| | EM | F1 | EM | F1 |
| w/o Evidence | 22.44 | 34.54 | 19.78 | 36.54 |
| Retrieve&Summary | 38.00 | 47.82 | 27.26 | 43.13 |
| GENREAD | 37.98 | 49.47 | 28.20 | 45.49 |

Table 2: Ablation study results on NQ and WebQ.

| Method | NQ | TriviaQA | WebQ |
|--------|-----|----------|------|
| Retrieve-Then-Answer | 59.2% | 64.6% | 70.0% |
| ALLIES | 69.6% | 72.9% | 82.0% |

Table 3: Query complementation analysis.

(2) Among the methods that do not use a retriever, GENREAD achieves the highest performance. This demonstrates the effectiveness of the generate-then-read pipeline, where the model generates background documents based on its own knowledge without relying on external corpus.

(3) Our implemented baselines, such as MCR and self-Ask, may not perform as well as expected. This is mainly because these methods heavily rely on result parsing, which limits their generalizability to other applications.

(4) Our proposed method, ALLIES, outperforms all existing baselines and achieves the highest performance on all datasets. This confirms the effectiveness of our model and demonstrates its superiority in open-domain question answering tasks. Additionally, our method relies less on result parsing, making it more generalizable to other applications.

## 5.5 Ablation Study

In ALLIES, we utilize LLMs to ask questions and retrieve evidence based on those questions. To investigate the effects of the evidence, we conduct

ablations by removing the evidence and using different types of evidence, as shown in Table 2.

Based on the results, we draw several conclusions: (1) When the evidence is removed, we only provide the LLM with related queries without any background information. In this case, the model's performance drops significantly, which confirms that incorporating evidence into the model can greatly improve its understanding of the query. (2) When using the LLM-generated background document (GENREAD), we observe that our model achieves slightly better results compared to retrieval & summary. This finding aligns with the observations made in GENREAD [Yu et al., 2022]. The improved performance can be attributed to the fact that LLMs have seen these related documents during pretraining, and the generated documents are more specific and refined.

## 5.6 Query Complementation Analysis

By iteratively generating new queries to complement the original query, our ALLIES is capable of expanding the information coverage of the original query and capturing hidden knowledge that may not be directly obtainable through retrieval with

| Method | Retrieval Times | API Times | Tokens Per API | Tokens Per Query |
|---|---|---|---|---|
| Directly Answer | 0 | 1 | 54 | $1 \times 54 = 54$ |
| GENREAD [Yu et al., 2022] | 0 | 1 | 342 | $1 \times 342 = 342$ |
| Self-Ask [Press et al., 2022] | 0 | 1 | 490 | $1 \times 490 = 490$ |
| Retrieve-Then-Answer (Top-5) | 1 | 1 | 744 | $1 \times 744 = 744$ |
| ALLIES (GENREAD) | 0 | 19 | 290 | $19 \times 290 = 5510$ |
| ALLIES (Retrieval&Summary) | 5 | 19 | 352 | $19 \times 352 = 6688$ |
| MCR [Yoran et al., 2023] | 12 | 12 | 3029 | $12 \times 3029 = 36348$ |

Table 4: The effectiveness analysis of ALLIES.

the original query. To verify this, we conduct a query complementation analysis that compares the retrieval results of retrieve-then-answer and AL-LIES. Specifically, we record the percentage of retrieval results containing the ground truth answer and present the findings in Table 3.

From the result, we can find that the retrieval results of ALLIES outperform those of retrieve-then-answer across all datasets, which verifies the effectiveness of ALLIES. By iteratively generating new queries, we can expand the knowledge scope of the retrieval results, leading to a more comprehensive understanding of the original query and naturally producing better answers.

## 5.7 Effectiveness Analysis

In ALLIES, the use of multiple iterations of retrieval and generation may introduce additional costs. To analyze its effectiveness, we utilize the complete set of questions from the NQ dataset to conduct the effectiveness analysis, which systematically compares the effectiveness of several methods.

As shown in Table 4, we can have the following conclusions: (1) Multi-turn QA methods, including ALLIES and MCR, incur higher model inference costs compared to single-turn QA methods such as Directly Answer, GENREAD, Self-Ask, and Retrieve-Then-Answer. This increase in cost is primarily due to the multiple API calls involved. (2) Among the multi-turn QA methods, although ALLIES requires more API calls, the token consumption per API is significantly lower than that of MCR, resulting in $1/6$ inference cost of MCR. The higher token consumption per API in MCR can be attributed to the demonstration, which consumes a substantial number of tokens. (3) Generally, single-turn QA methods have lower token costs but exhibit lower model performance. In contrast, ALLIES achieves significantly better model performance while maintaining an acceptable token cost compared to MCR, thus demonstrating the effectiveness of our method.

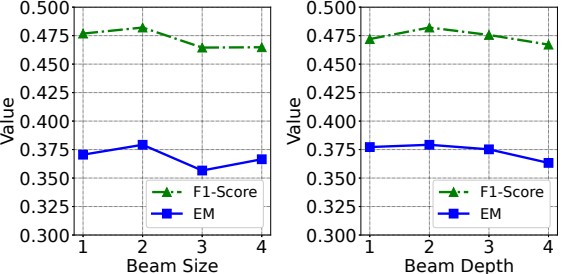

Figure 3: Performance comparison *w.r.t.* hyper-parameters on NQ dataset.

## 5.8 Human Evaluation

In this section, we conducted a human evaluation to assess the accuracy of the scores generated by LLMs in our scoring function. We randomly selected 100 samples for score calculation and manually verified the generated scores.

Our findings indicate that 93 percent of the generated scores align with the requirements for score calculation. This validation confirms the rationale behind using LLMs to calculate the scores. However, we also observed some rare cases where two answers could both potentially address the question, but one of them was more accurate. In these cases, the LLMs assigned the same score to both answers, potentially leading to the selection of the less accurate answer. This issue can be attributed to the coarse nature of the prompt used for scoring, which can only assess the general relevance score. To address this issue, one possible solution for future work is to calculate the scores using an ensemble-and-vote approach. This would involve asking LLMs to rank all possible answers instead of scoring them individually, which would potentially achieve more accurate and reliable scores.

## 5.9 Hyper-parameter Study

Beam size $B$ and beam depth $D$ are two important hyper-parameters in our method. We study their effects by changing one parameter while fixing

| | |
|---|---|
| **Question:** Who led the soldiers in ending the raid on the harper's ferry arsenal? | |
| **Answer:** [Brevet Colonel Robert E. Lee,First Lieutenant Israel Greene] | |
| **Generated Query:** | |
| - What was the name of the leader who led the soldiers in ending the raid on the Harper's Ferry arsenal? | |
| - Who was the overall commander of the soldiers who led the operation to retake the arsenal at Harpers Ferry? | |
| **Retrieved Evidence:** | |
| - The soldiers who led the operation to retake the arsenal at Harpers Ferry were under the overall command of Colonel Robert E. Lee. | |
| - Colonel Robert E. Lee was in overall command of the operation to retake the arsenal. It is possible that he may have played a role in leading the soldiers to end the raid. | |
| **Generated Answer with Score:** | |
| - Answer 1: Colonel Robert E. Lee. Score: 0.8 ✔      - Answer 2: First Lieutenant Israel Greene. Score: 0.7 | |
| **Model Predictions:** | |
| Directly Answer: John Brown     Retrieve-Then-Answer: John Brown     ALLIES: Colonel Robert E. Lee ✔ | |
| **Question:** When was the first driver's license required? | |
| **Answer:** 1 January 1904 | |
| **Generated Query:** | |
| - In which country was the first driver's license required? | |
| - When did the UK implement mandatory licensing for drivers and what was the minimum qualifying age? | |
| **Retrieved Evidence:** | |
| - The first driver's license requirement was mandated on January 1, 1904, in the United Kingdom after the Motor Car Act 1903 received royal assent. The minimum qualifying age was set at 17, and every car owner... | |
| - The first formal driving test in the UK was introduced with the Road Traffic Act 1934, which made compulsory testing for all new drivers. Prior to this, UK driving licenses were introduced by the Motor Car Act 1903... | |
| **Generated Answer with Score:** | |
| - Answer 1: January 1, 1904. Score: 0.9 ✔      - Answer 2: 1903. Score: 0.8 | |
| **Model Predictions:** | |
| Directly Answer: 1903     Retrieve-Then-Answer: July 1913     ALLIES: 1 January 1904 ✔ | |

Table 5: Case studies of the process of ALLIES.

other parameters and observing the performance trends, which are shown in Figure 3.

**Study on Beam Size** $B$**.** Beam size refers to the number of questions we keep at each layer during answer searching. From the results, we observe that the performance reaches its peak when the beam size ($B$) is set to 2. Values smaller or larger than this threshold lead to performance degradation. This is primarily because a larger beam size provides the model with more opportunities to make mistakes. However, when the beam size is too large, the model struggles to effectively rank the multiple candidates and select the best answer. Additionally, an increase in beam size also incurs additional computational costs.

**Study on Beam Depth** $D$**.** Beam depth refers to the maximum depth our model can reach during answer searching. From the results, we find that the performance change during beam depth tuning is relatively small. This is mainly due to the early stop mechanism we implemented, where the answer searching can terminate before reaching the maximum search depth if the answer score surpasses the threshold. However, we also observe that when the beam depth is too large (e.g., 4), the model's performance starts to decline. We be-

lieve this is mainly because, in most cases, a beam depth of 2 provides the model with sufficient background information. Increasing the beam depth beyond that only introduces more noisy information, which may complicate the generation of the correct answer for the LLM.

### 5.10 Case Study

In this section, we provide examples that illustrate the reasoning process of our ALLIES method, which is shown in Table 5. From these examples, we draw the following conclusions:

(1) The generated queries in our method are more specific and focused compared to the original query. This specificity improves the accuracy of the retrieval process, resulting in more accurate and relevant retrieved evidence. Consequently, the generated answers are of higher quality.

(2) During the answer generation process, there might be instances where wrong answers are initially predicted. However, our scoring function effectively assigns lower scores to these wrong answers based on the reasoning history. As a result, the final output is the correct answer. This demonstrates the robustness of our method in handling potential mistakes and effectively filtering out incorrect answers.

## 6   Conclusion

In this paper, we introduce ALLIES, a novel method that addresses the limitations of using large language models (LLMs) for complex tasks. By leveraging LLMs to generate related queries iteratively, ALLIES enables iterative reasoning and expands the original query's scope to capture hidden knowledge. We evaluate ALLIES in zero-shot open-domain question answering and demonstrate its superiority over other baselines on benchmarks. As for future work, we plan to apply ALLIES in other complex tasks such as mathematical reasoning and so on.

## Limitations

In this work, we propose an effective response generation method ALLIES. The limitations of the proposed method are as follows:

(1) The computational cost of ALLIES is relatively high due to the need for multiple API calls and document retrieval. This can limit its practicality in resource-intensive scenarios or systems with limited computational resources.

(2) The operation of the model is based on the designed prompt. When applied to a new application scenario, crafting effective prompts may require additional time and effort from users.

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

## A    Data Statistics

The statistics of used datasets are shown in Table 8.

## B    Hyper-parameters

The detailed hyper-parameters are shown in Table 6.

## C    Detailed Prompts of the Functions

### C.1    Answering Function $\text{Answer}(q, \mathcal{Q}, \mathcal{E})$

Given the following query-evidence pair:
{query-evidence pair}
Please refer to the query-evidence pair above, answer the following question with just one entity.
Question: {query}
The answer is:

### C.2    Asking Function $\text{Ask}(q, \mathcal{Q}, \mathcal{E}, K)$

Given the question:
{query}
and following query-evidence pair:
{query-evidence pair}.
Please generate some questions that can help answer the given question with the following constraints:
1. You should output no more than $k$ questions.
2. You should directly output the ranked sub-questions based on their importance.
3. The generated questions should be diverse and focus on different aspects of the given question.
4. You should output in the following format:
Ranked Questions:
1. [Question 1] ...

### C.3    Retrieval Function $\text{Retrieve}(q_{ori}, q, N)$

Given the original question:
{query}
and the provided document:
{doc}
output the factual information from the evidence that is relevant to the question:

### C.4    Retrieval Function $\text{Retrieve}'(q_{ori})$

Generate a short background document from Wikipedia to answer the given question: {query}

### C.5    Scoring Function $\text{Score}(q, \mathcal{Q}, \mathcal{E}, a)$

Given the question:
{query}
and the candidate answer:
{answer}
and the Query-evidence pair:
{query-evidence pair}
refer to the query-evidence pair below and utilize your own reasoning ability to assess the probability that the candidate answer is the true answer.
Please provide a number between 0 and 1 as the output, following the guidelines below:
If the probability is between 0 and 0.3, it signifies that the model has substantial evidence to suggest it is an incorrect answer.
If the probability is between 0.3 and 0.5, it suggests that the model leans towards considering it an incorrect answer, but lacks concrete evidence.
If the probability is between 0.5 and 0.7, it indicates that the model leans towards considering it a correct answer, but lacks concrete evidence.
If the probability is greater than 0.7, it signifies that the model has substantial evidence to suggest it is the correct answer.
If the candidate answer doesn't provide clear solution to the question, the probability should be 0.
The score is:

## D    Dense Retriever

**Dual Encoder.**    The predominant architecture currently utilized for dense retrieval is known as the dual encoder. This architecture employs dense vector representations, denoted as $\boldsymbol{q}$ and $\boldsymbol{d}$, to encode queries and documents, respectively. The similarity scores are then computed using the inner product as follows:

$$s(\boldsymbol{q}, \boldsymbol{d}) = E_Q(\boldsymbol{q})^T \cdot E_D(\boldsymbol{d}) \qquad (1)$$

where $E_Q(\cdot)$ and $E_D(\cdot)$ refer to the query encoder and document encoder, respectively. To leverage the embeddings, existing solutions typically employ approximate nearest neighbor (ANN) search algorithms such as FAISS [Johnson et al., 2021].

**Performance of Dual Encoder.**    The pre-trained language model (PLM) used in the training of retrievers is COCONDENSER[2]. The performances of DEs on different datasets can be found in Table 7.

---

[2]Luyu/co-condenser-marco in huggingface.

| Parameter | NQ | TriviaQA | WebQ |
|---|---|---|---|
| Threshold | 0.8 | 0.8 | 0.8 |
| Beam Size | 2 | 3 | 3 |
| Beam Depth | 2 | 1 | 2 |
| Retrieval Number | 2 | - | - |
| Expand Question Number | 2 | 2 | 3 |
| Evidence Type | Retrieval | GENREAD | GENREAD |
| LLM API | GPT-3.5-Turbo | GPT-3.5-Turbo | GPT-3.5-Turbo |

Table 6: Hyper-parameters for ALLIES.

| Dataset | R@1 | R@5 | R@20 | R@50 | R@100 | R@1k | MRR@10 | MAP@1k |
|---|---|---|---|---|---|---|---|---|
| NQ | 46.43 | 68.86 | 80.28 | 84.40 | 86.86 | 92.06 | 56.03 | 21.96 |
| TriviaQA | 58.34 | 73.44 | 80.71 | 84.04 | 85.95 | 89.55 | 64.71 | 25.03 |
| WebQ | 52.31 | 72.10 | 80.41 | 83.76 | 85.63 | 90.80 | 60.72 | 21.50 |

Table 7: The results of the dual encoders on different datasets.

| Datasets | Train | Valid | Test |
|---|---|---|---|
| NQ [Kwiatkowski et al., 2019] | 79,168 | 8,757 | 3,610 |
| TriviaQA [Joshi et al., 2017] | 78,785 | 8,837 | 11,313 |
| WebQ [Berant et al., 2013] | 3,478 | 300 | 2,032 |

Table 8: Datasets splits and statistics.