# OpenReview forum: "Allies: Prompting Large Language Model with Beam Search"
_EMNLP/2023/Conference — EMNLP 2023 Findings_

### Official Review · Reviewer_7GwV · 2023-07-26

**Soundness:** 4

**Excitement:**

3: Ambivalent: It has merits (e.g., it reports state-of-the-art results, the idea is nice), but there are key weaknesses (e.g., it describes incremental work), and it can significantly benefit from another round of revision. However, I won't object to accepting it if my co-reviewers champion it.

**Paper Topic And Main Contributions:**

This paper marks the introduction of ALLIES, an innovative approach that leverages a beam search strategy for response generation in open-domain question answering. With a unique extension strategy and dynamic pruning technique, ALLIES capably tackles the critical limitations of narrow information coverage and low fault tolerance associated with large language models. Demonstrated on popular benchmarks, ALLIES excels, significantly outperforming existing methods.

Overall, the contribution of the paper is substantial, leading me to lean towards accepting it.

**Questions For The Authors:**

- Concerning the issue of narrow information coverage, how do the authors view the relevance of their work in the context of query expansion [1]?
- In relation to the low fault tolerance, how do the authors view the relevance of their work with self-consistency [2]?
- Have the authors experimented with varying prompts at each step to assess prompt robustness?
- In Figure 3, the paper presents the sensitivity of the beam size and depth. Could the authors provide their insights as to why an increase in depth leads to diminished performance?

[1] Generation-augmented retrieval for open-domain question answering. ACL 2021
[2] Self-consistency improves chain of thought reasoning in language models. ICLR 2023

**Reasons To Accept:**

- The paper effectively motivates its work by addressing the prevalent challenges of narrow information coverage and low fault tolerance encountered in existing methods utilizing LLMs for query responses.
- The proposed ALLIES method is refreshingly innovative, employing an extension strategy for expansive information coverage and utilizing a dynamic pruning technique to maintain top-tier responses, thereby fostering the model's capacity to accommodate inaccuracies and alternative answers. This approach enhances response precision and reliability.
- The paper successfully demonstrates the superiority of ALLIES over established methods through comprehensive experiments on zero-shot open-domain QA benchmarks, including NQ, TriviaQA, and WebQ, exhibiting marked improvements (+10 on NQ and WebQ).

**Reasons To Reject:**

- There is a noticeable increase in computational costs due to the proposed method's iterative nature, encompassing the retrieval and generation process.
- While the paper illustrates its limitations in the introduction, the authors fail to provide a comprehensive comparison between ALLIES and existing methods, leaving the contribution of this work less well-defined.
- Despite the use of a single prompt in the paper, the sensitivity of the model to variations in the prompt was not evaluated, leaving an important aspect of the method unexplored.

**Reproducibility:**

4: Could mostly reproduce the results, but there may be some variation because of sample variance or minor variations in their interpretation of the protocol or method.

**Reviewer Confidence:**

4: Quite sure. I tried to check the important points carefully. It's unlikely, though conceivable, that I missed something that should affect my ratings.

---

> ### Author Rebuttal · Authors · 2023-08-28
>
> We deeply appreciate your positive feedback. Here are the addressed points for your review:
>
> 1. **Regarding the Distinction Between Our Work and GAR**
>
>    The distinctions between our work and GAR are delineated as follows:
>
>    - GAR emphasizes general semantic augmentation of the original query, while ALLIES concentrates on identifying gaps in LLM's knowledge and generating new queries to extend its knowledge boundaries.
>    - Unlike GAR, ALLIES-generated queries might be less related to the original query but significantly contribute to broadening the knowledge scope of the retrieved evidence.
>
>    A comparison with GAR will be integrated into the upcoming version of our paper.
>
> 2. **Regarding the Distinction Between Our Work and Self-Consistency**
>
>    The distinctions between our work and Self-Consistency are delineated as follows:
>
>    - Self-Consistency generates multiple answers based on the same evidence input, while ALLIES generates answers considering various evidence inputs, including the previous reasoning process.
>    - While both methods enhance fault tolerance, ALLIES provides LLM with more evidence during reasoning, potentially yielding improved performance.
>
>    It's noteworthy that Self-Consistency could be integrated into our framework by allowing LLM to generate multiple answers in the Answering Function. A comparison with Self-Consistency will also be included in the future version of the paper.
>
> 3. **Regarding the Prompt Choice**
>
>    We conducted experiments with five different prompts for each step and found that distinct prompts yielded similar results.
>
> 4. **Regarding the Performance Degradation with Increased Beam Depth**
>
>    Upon meticulous analysis, we attribute the observed performance degradation when increasing the beam depth to the following reasons:
>
>    - Most questions can be answered within a beam depth of 2. Elevating the beam depth tends not to improve the results for these questions.
>    - For particularly challenging questions, heightened beam depth might introduce more noise, potentially leading to misleading information and performance degradation.
>
> We deeply appreciate your positive feedback. If these clarifications align with your expectations, we kindly ask for your consideration in **raising the Excitement score**. Your feedback significantly contributes to the advancement of our work.

---

### Official Review · Reviewer_KsFN · 2023-08-05

**Soundness:** 3

**Excitement:**

4: Strong: This paper deepens the understanding of some phenomenon or lowers the barriers to an existing research direction.

**Paper Topic And Main Contributions:**

The paper introduces Allies, a novel method designed to address the limitations of using large language models (LLMs) for complex tasks. Allies leverages LLMs to generate related queries iteratively, enabling iterative reasoning and expanding the scope of the original query to capture hidden knowledge. The proposed method is evaluated in zero-shot open-domain question answering and outperforms other baselines on benchmark datasets.

**Questions For The Authors:**

1. How do beam depths larger than 3 impact the results? It would be beneficial to provide specific examples illustrating the effects of longer reasoning chains.
2. With a larger beam size, can we expect the generation of more noisy or greater numbers of augmented queries?

**Reasons To Accept:**

1. The paper identifies issues with stacking LLM queries and proposes a new framework to optimize iterative calling of LLM APIs.
2. Allies presents a robust method for achieving prompt augmentation, enhancing the model's performance.
3. Experimental results demonstrate the superior performance of Allies on open book QA benchmarks.
4. The paper introduces a valuable approach for addressing the limitations of using LLMs for complex tasks.

**Reasons To Reject:**

1. The limitations of beam depths are not thoroughly explored. Further investigation is needed to understand how longer reasoning chains affect performance.
2. The potential impact of larger beam sizes could be explored further.

**Reproducibility:**

3: Could reproduce the results with some difficulty. The settings of parameters are underspecified or subjectively determined; the training/evaluation data are not widely available.

**Reviewer Confidence:**

4: Quite sure. I tried to check the important points carefully. It's unlikely, though conceivable, that I missed something that should affect my ratings.

---

> ### Author Rebuttal · Authors · 2023-08-28
>
> We are grateful for your positive feedback. Please allow us to address your questions as follows:
>
> 1. **Regarding the Performance of Beam Depth Larger than 3**
>
>    In our experimentation, as outlined in Section 5.9, we examined the impact of varying the beam depth ranging from 1 to 4. During the rebuttal phase, we expanded the beam depth to 6, and the corresponding results are provided in the following table:
>
>    | Beam Depth |   1    |     2      |   3    |   4    |   5    |   6    |
>    | ---------- | :----: | :--------: | :----: | :----: | :----: | :----: |
>    | EM         | 0.3772 | **0.3792** | 0.3752 | 0.3633 | 0.3669 | 0.3631 |
>    | F1-Score   | 0.4720 | **0.4821** | 0.4756 | 0.4671 | 0.4692 | 0.4663 |
>
>    The observed trend indicates that performance declines as beam depth increases beyond 2. Our rationale for this outcome lies in the following facts:
>
>    - Most questions can be effectively addressed within a beam depth of 2, and raising the depth does not necessarily yield improved outcomes for these questions.
>    - For particularly challenging questions, a heightened beam depth might introduce more noise, potentially complicating the LLM's ability to generate accurate responses.
>
>    In our upcoming paper version, we intend to provide specific examples that vividly illustrate the impact of extended reasoning chains.
>
> 2. **Regarding the Impact of a Larger Beam Size**
>
>    In Section 5.9, we conducted an experiment where we adjusted the beam size within the range of 1 to 4. During the rebuttal phase, we expanded the beam size to 6, and the corresponding results are presented in the following table:
>
>    | Beam Size |   1    |     2      |   3    |   4    |   5    |   6    |
>    | --------- | :----: | :--------: | :----: | :----: | :----: | :----: |
>    | EM        | 0.3705 | **0.3792** | 0.3566 | 0.3665 | 0.3745 | 0.3721 |
>    | F1-Score  | 0.4768 | **0.4821** | 0.4645 | 0.4648 | 0.4743 | 0.4605 |
>
>    The results illustrate that our model's optimal performance is achieved with a beam size of 2. Values both below and above this threshold result in performance deterioration. This phenomenon can be explained by:
>
>    - A larger beam size provides the model with a greater variety of augmented queries and an increased ability to accommodate mistakes, potentially leading to improved performance.
>    - However, upon careful analysis, we noticed that the queries generated by beam size larger than 2 exhibit substantial similarity, yielding minimal additional information. Consequently, the model encounters challenges in effectively ranking multiple candidates and identifying the optimal answer, leading to a decline in performance.
>
> 3. **Regarding the Reproducibility Issue**
>
>    We have included our hyperparameter settings, detailed prompts, and comprehensive details about our dense retrievers in the Appendix and we will open source the code upon acceptance.
>
> We are grateful for your positive feedback. If these responses align with your expectations, we kindly request your consideration in **raising the Soundness score**. Your engagement contributes significantly to the advancement of our work.

---

### Official Review · Reviewer_rUD4 · 2023-08-12

**Soundness:** 3

**Excitement:**

3: Ambivalent: It has merits (e.g., it reports state-of-the-art results, the idea is nice), but there are key weaknesses (e.g., it describes incremental work), and it can significantly benefit from another round of revision. However, I won't object to accepting it if my co-reviewers champion it.

**Paper Topic And Main Contributions:**

For open domain question answering, this paper employed an extension strategy to broaden the scope of the original question by generating multiple relevant questions, which allows the LLM to develop a more comprehensive understanding of the complex question by examining its individual components. Throughout the iterative process, a dynamic pruning technique was utilized to narrow down the options, retaining only the top answers at each step.

**Questions For The Authors:**

Question A: What are the distinctions between the interactive and iterative process depicted in Figure 1 and the algorithm based on a chain of thought?

Question B: While beam search is not a novel technique, what is the main contribution of its utilization in this paper?

Question C: The authors state that InstructGPT is employed, indicating LLM alignment with human feedbacks. How is beam search implemented on the top InstructGPT?

Question D: Does ALLIES provide an answer based solely on the top-1 retrieved evidence?

Question E: The authors mention scoring the response using the LLM. How is the score qualified or evaluated?

**Reasons To Accept:**

The testing results appear favorable.

**Reasons To Reject:**

The major contributions do not exhibit significant impact.  One instance is the utilization of beam search, which is not a new technique. Therefore, it is crucial to identify the main contribution of using beam search in this study. Additionally, it is important to elucidate the distinctions between the interactive and iterative process and the chain of thought approach.

**Reproducibility:**

4: Could mostly reproduce the results, but there may be some variation because of sample variance or minor variations in their interpretation of the protocol or method.

**Reviewer Confidence:**

4: Quite sure. I tried to check the important points carefully. It's unlikely, though conceivable, that I missed something that should affect my ratings.

**Typos Grammar Style And Presentation Improvements:**

It is hard to follow the "Algorithm 1 The process of generating the response to a given query using ALLIES" on page 4.

---

> ### Author Rebuttal · Authors · 2023-08-28
>
> Thank you for your insightful feedback. We have taken your inquiries into careful consideration and would like to provide detailed clarifications:
>
> 1. **Regarding the Differences from Chain of Thought (CoT)**
>
>    We extend our apologies for any confusion caused. To clarify the distinctions between our algorithm and Chain of Thought, we highlight the following aspects:
>
>    - CoT employs an iterative chain approach, potentially exhibiting low fault tolerance. Conversely, our algorithm adopts an expand-and-iterative tree approach that permits LLM to make mistakes while answering questions.
>    - While most CoT-based methods rely on demonstrations to complement their prompts, our approach doesn't incorporate any demonstrations.
>
>    Notably, one of our baselines, Self-Ask[1], follows a CoT-style approach to answering questions. However, empirical results demonstrate its inferior performance compared to our method, despite incurring significantly higher computational costs. A detailed comparison will be included in the forthcoming version of our paper.
>
> 2. **Regarding the Contributions of the Utilization of Beam Search**
>
>    The contributions of utilizing beam search in our work can be concisely outlined as follows:
>
>    - During Beam Expansion, additional queries are generated iteratively, driven by the LLM's inquiries about the required information. This strategy enhances the breadth of information coverage of the LLM.
>    - During Beam Pruning, we retain the top B highest-scoring answers, affording the LLM the capacity to make erroneous predictions, thereby bolstering the model's fault tolerance.
>
> 3. **Regarding the Implementation of Beam Search on InstructGPT**
>
>    As stated in Section 3 and Algorithm 1, our beam search implementation unfolds as follows:
>
>    - Beam Initialization: LLM directly answers the question and generates answers based on evidence, respectively.
>    - Beam Expansion: LLM generates new queries iteratively, broadening information coverage.
>    - Beam Pruning: LLM scores generated answers, retaining the highest-scoring B answers.
>    - Beam Termination: Terminate the beam reasoning when answer scores exceed a threshold or maximum depth is reached.
>
> 4. **Regarding the Number of Retrieved Evidence**
>
>    Detailed hyperparameters concerning the number of retrieved evidence are illustrated in Table 6 of the Appendix section. It's crucial to highlight that ALLIES supports diverse evidence types. Specifically, for NQ, the retriever retrieves the top-2 passages as evidence, whereas for TriviaQA and WebQ, GENREAD generates a background document serving as evidence.
>
> 5. **Regarding the Evaluation of LLM-Generated Scores**
>
>    As detailed in Section 5.8, we conducted score calculations using 100 randomly selected samples, subjecting the generated scores to manual verification. Our findings indicate a 93 percent alignment between the generated scores and the criteria for score calculation.
>
> Furthermore, we are committed to facilitating a clearer understanding by incorporating additional illustrative examples throughout our paper.
>
> We are sincerely grateful for your valuable feedback. If these clarifications meet your expectations, we kindly request your consideration in **raising the Soundness score**.
>
> **References:**
>
> [1] Press O, Zhang M, Min S, et al. Measuring and narrowing the compositionality gap in language models[J]. arXiv preprint arXiv:2210.03350, 2022.

---

### Official Review · Reviewer_2w8K · 2023-08-13

**Soundness:** 3

**Excitement:**

3: Ambivalent: It has merits (e.g., it reports state-of-the-art results, the idea is nice), but there are key weaknesses (e.g., it describes incremental work), and it can significantly benefit from another round of revision. However, I won't object to accepting it if my co-reviewers champion it.

**Paper Topic And Main Contributions:**

The paper introduces an iterative approach to enhance the retrieval-augmented generation (RAG) pipeline through query augmentation.
The method involves leveraging Language Models (LLMs) to generate additional related queries in an iterative manner, and then scoring the output using both the query and the retrieved evidence.
The effectiveness of the proposed approach is demonstrated through a comparison with the retrieve-then-read and generate-then-read pipelines.

**Questions For The Authors:**

1. What is the corpus for the retrieval?

**Reasons To Accept:**

Iterative query augmentation with scoring function can improve the RAG pipeline significantly.

**Reasons To Reject:**

While the paper demonstrates its effectiveness, there are some minor concerns that I would like to address:

1. Lack of experiment details:
- Providing explicit details regarding the corpus or external sources employed for retrieval purposes would be helpful
- Table 4 could benefit from further explanation to improve understanding.
    - It seems that the API time for GENREAD should be listed as 2 instead of 1.
    - It is unclear which specific example was utilized to generate the results presented in Table 4. Does the "retrieval&summary" category for ALLIES indicate that the process was repeated five times?

2. Unclear why WebQ results show performance decline with additional context which is contrast behavior with previous findings.

**Reproducibility:**

4: Could mostly reproduce the results, but there may be some variation because of sample variance or minor variations in their interpretation of the protocol or method.

**Reviewer Confidence:**

4: Quite sure. I tried to check the important points carefully. It's unlikely, though conceivable, that I missed something that should affect my ratings.

---

> ### Author Rebuttal · Authors · 2023-08-28
>
> Thank you for your valuable feedback. We would like to address your inquiries as follows:
>
> 1. **Regarding the Corpus Source and Details**
>
>    We apologize for any oversight in providing complete details. Following [1,2], we use the Wikipedia dump from Dec. 20, 2018 as our retrieval corpus, encompassing a comprehensive collection of 21,015,324 documents. We will provide the corpus download link in the future version of the paper.
>
> 2. **Regarding the API Times for GENREAD**
>
>    We extend our gratitude for identifying the discrepancy in the API times for GENREAD. We acknowledge this error and are committed to rectifying it in the future version of the paper.
>
> 3. **Regarding the API Times for ALLIES (Retrieval&Summary)**
>
>    We deeply regret any confusion that may have arisen. In our experimentation, we utilized the complete set of questions from the NQ dataset to conduct the effectiveness analysis. It was observed that, on average, ALLIES requires 4.7 retrieval times to address a single question. For the sake of clarity and ease of comprehension, we have rounded this figure to 5. It is our intention to provide a more detailed illustration of this process in future versions of the paper.
>
> 4. **Regarding WebQ's Performance Decline with Additional Context**
>
>    During our experimentation, we observed the trend of diminished performance on WebQ with the introduction of supplementary context. Following meticulous analysis, we have identified two principal explanations for this phenomenon:
>
>    - **False Positive Documents:** The retrieval process might yield documents that are not entirely relevant to the given question, as also acknowledged in [3]. Consequently, this introduces the presence of false positive documents, which in turn introduces noise information that may adversely affect the overall model performance.
>    - **Time-Dependent Questions:** Introduced in 2013, WebQ includes inherently time-dependent questions, featuring answers that may have become outdated over time. Contrastingly, the Wikipedia corpus (from Dec. 20, 2018) we employ is substantially more current. This temporal misalignment leads to discrepancies between the LLM's generation of new answers and the pre-existing answers within the WebQ dataset, which particularly impacts the EM and F1-Score metrics. This phenomenon of time-dependent answers is also explored in [2].
>
> 5. **Regarding the Reproducibility Issue**
>
>    We have included our hyperparameter settings, detailed prompts, and comprehensive details about our dense retrievers in the Appendix. We will provide the corpus download link in the future version of the paper and open-source the code upon acceptance.
>
> We greatly value your insightful feedback. If these clarifications meet your expectations, we kindly request your consideration in **raising the Soundness score and Reproducibility score**.
>
> **References:**
>
> [1] Karpukhin V, Oğuz B, Min S, et al. Dense passage retrieval for open-domain question answering[J]. arXiv preprint arXiv:2004.04906, 2020.
>
> [2] Yu W, Iter D, Wang S, et al. Generate rather than retrieve: Large language models are strong context generators[J]. arXiv preprint arXiv:2209.10063, 2022.
>
> [3] Singh Sachan D, Lewis M, Yogatama D, et al. Questions Are All You Need to Train a Dense Passage Retriever[J]. arXiv e-prints, 2022: arXiv: 2206.10658.

---

### Meta-Review · Area_Chair_6a33 · 2023-09-21

**Recommendation:** 3

**Metareview:**

Allies is a method for iteratively querying an LLM in a beam-search style. Specifically, they
- use a query expansion method that retrieves relevant data and adds it to the prompt
- for each retrieved expansion, score it and prune the low-scoring ones
- repeat the above two steps until the score is > a certain threshold

Expansion followed by pruning enables augmenting with missing information + being tolerant to wrong retrievals. The retriever is fine-tuned. The answerer and the scorer are both ChatGPT3.5-turbo.

The method is in the same category as methods like self-ask, retrieve-then-ask, and GENREAD. It has been implemented and evaluated on open-domain QA for the datasets NQ, TriviaQA, and WebQ. Results are consistently better compared to relevant baselines.

Reviewers 1 and 2 had complaints about paper writing, which can be addressed in the camera-ready version.
Reviewers 3 and 4 questions have been answered.

The method requires ~10x increase in number of API calls compared to some of the baselines in the same category. While expensive, it actually makes at interesting point about the fact that better results can be achieved if the user is willing to call the API many more times.

Overall, the approach makes sense and the results are pretty positive.

---

### Decision · Program_Chairs · 2023-10-07

**Decision:**

Accept-Findings

**Comment:**

Allies is a method for iteratively querying an LLM in a beam-search style. Specifically, they
- use a query expansion method that retrieves relevant data and adds it to the prompt
- for each retrieved expansion, score it and prune the low-scoring ones
- repeat the above two steps until the score is > a certain threshold

Expansion followed by pruning enables augmenting with missing information + being tolerant to wrong retrievals. The retriever is fine-tuned. The answerer and the scorer are both ChatGPT3.5-turbo.

The method is in the same category as methods like self-ask, retrieve-then-ask, and GENREAD. It has been implemented and evaluated on open-domain QA for the datasets NQ, TriviaQA, and WebQ. Results are consistently better compared to relevant baselines.

Reviewers 1 and 2 had complaints about paper writing, which can be addressed in the camera-ready version.
Reviewers 3 and 4 questions have been answered.

The method requires ~10x increase in number of API calls compared to some of the baselines in the same category. While expensive, it actually makes at interesting point about the fact that better results can be achieved if the user is willing to call the API many more times.

Overall, the approach makes sense and the results are pretty positive.